# Enhanced NOMA System Using Adaptive Coding and Modulation Based on LSTM Neural Network Channel Estimation

**Mai AbdelMoniem [1,\*], Safa M. Gasser [1], Mohamed S. El-Mahallawy [1] 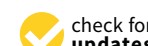,
Mohamed Waleed Fakhr [2] and Abdelhamid Soliman [3]**

[1] Department of Electronics and Communications Engineering, Arab Academy for Science, Technology and Maritime Transport (AASTMT), Cairo P.O. Box 2033, Egypt; safagasser@aast.edu (S.M.G.); mahallawy@aast.edu (M.S.E.-M.)

[2] Department of Computer Engineering, Arab Academy for Science, Technology and Maritime Transport (AASTMT), Cairo P.O. Box 2033, Egypt; waleedf@aast.edu

[3] School of Creative Arts and Engineering, Staffordshire University, Stoke-on-Trent ST4 2DE, UK; A.Soliman@staffs.ac.uk

\* Correspondence: mai.abdelmoniem@aast.edu

**Abstract:** Non-orthogonal multiple access (NOMA) is the technique proposed for multiple access in the fifth generation (5G) cellular network. In NOMA, different users are allocated different power levels and are served using the same time/frequency resource blocks (RBs). The main challenges in existing NOMA systems are the limited channel feedback and the difficulty of merging it with advanced adaptive coding and modulation schemes. Unlike formerly proposed solutions, in this paper, we propose an effective channel estimation (CE) algorithm based on the long-short term memory (LSTM) neural network. The LSTM has the advantage of adapting dynamically to the behavior of the fluctuating channel state. On average, the use of LSTM results in a 10% lower outage probability and a 37% increase in the user sum rate as well as a maximal reduction in the bit error rate (BER) of 50% in comparison to the conventional NOMA system. Furthermore, we propose a novel power coefficient allocation algorithm based on binomial distribution and Pascal's triangle. This algorithm is used to divide power among N users according to each user's channel condition. In addition, we introduce adaptive code rates and rotated constellations with cyclic Q-delay in the quadri-phase shift keying (QPSK) and quadrature amplitude modulation (QAM) modulators. This modified modulation scheme overcomes channel fading effects and helps to restore the transmitted sequences with fewer errors. In addition to the initial LSTM stage, the added adaptive coding and modulation stages result in a 73% improvement in the BER in comparison to the conventional NOMA system.

**Keywords:** adaptive coding; adaptive modulation; channel estimation; constellation rotation; cyclic-q delay; long-short term memory (LSTM); machine learning; non-orthogonal multiple access (NOMA); recurrent neural networks (RNNs)

## 1. Introduction

Non-orthogonal multiple access (NOMA) addresses the increasing demand on resources in Fifth Generation (5G) networks by accommodating several users within the same resource block (RB), which improves the bandwidth efficiency compared with orthogonal multiple access (OMA) techniques [1], i.e., time division multiple access (TDMA), frequency division multiple access (FDMA), code division multiple access (CDMA), and orthogonal frequency division multiple access (OFDMA).

NOMA can exploit the existing resources more efficiently by opportunistically taking advantage of the users' precise channel conditions and serving several users with diverse quality of service (QoS) requirements. Due to the new power domain dimension, NOMA systems can be combined with present multiple access (MA) models [2]. The power-domain NOMA (PD-NOMA) is considered a strong candidate for use in future 5G networks. PD-NOMA guarantees that several users are served with the same resources using superposition coding (SC) methods at the transmitter and successive interference cancellation (SIC) at the receiver [3]. PD-NOMA challenges lie in the possibility of appropriately allocating power coefficients according to each user's channel condition as well as the opportunity to merge NOMA with adaptive coding and modulation schemes in order to enhance the system's capacity and user sum rates. In order to resolve these issues, complete knowledge of the channel conditions (channel state information (CSI)) of individual users is essential [4]. Machine learning algorithms can adapt to changes and estimate the channel conditions for each user, and hence, they are considered strong candidates for future radio networks [2].

Previous studies mainly focused on how the outage probability of the NOMA system changes under different user conditions, excluding the variation in the channel [5,6]. The authors of [5,7,8] investigated error performance, power and resource allocation algorithms, and capacity performance in NOMA to enhance the reliability and efficiency of NOMA systems compared to OMA. Recently, numerous innovative enhancements to NOMA were proposed, aiming to discover the superlative candidate for implementation in 5G systems [9,10]. The authors of [11–14] examined the performance of downlink transmission in NOMA. Similarly, the authors of [15] studied the outage performance and the matching problem along with power allocation and decoding order ranges of a downlink system scenario, where the transmitter aims to send independent data to the receivers with specific data rates under the effect of statistically generated CSI.

In [16], an analysis of the NOMA outage performance was performed under two conditions: The QoS requirement of the corresponding data rate of users, and the channel's signal-to-noise ratio (SNR) for each user. The developed analytical results proved that NOMA can achieve superior performance in terms of users' data rates. Thus, choosing the users' targeted data rates and allocated power affects the outage performance of NOMA. The channel exhibited small-scale Rayleigh fading and the power coefficients were limited to a single value per user. The authors [16] found that the performance of NOMA at low SNR values was insignificant. In [17], the authors investigated variations in the outage probability and throughput against variable SNR values and under the impact of imperfect CSI using the minimum mean square error (MMSE) for channel estimation, but the calculation and assignment of power coefficients was not explained. The authors of [17] calculated the outage probability at a fixed distance from the base station (BS) and proposed a channel estimation method that was not adaptive to changes in the channel's response.

The authors of [18] studied the performance of the NOMA system in both the uplink and downlink by comparing the spectral efficiency of NOMA with respect to OMA systems. The authors of [18] mainly predicted a relation between the user selection criteria and the distance from the serving base transceiver station (BTS), taking into consideration the spectral efficiency of the system under an invariant channel. The authors of [19] aimed to maximize the sum rate in visible light communication (VLC) downlink NOMA systems by proposing a fair, non-complex optimum power allocation algorithm. In [20], the authors determined a closed-form expression for the optimum power allocation algorithm based on maximizing the sum rates in a single-input-single-output (SISO) downlink NOMA system. The authors proved that to reach the optimum solution, each user is required to decode the messages of users with lower SNRs first. The authors in [18] did not propose any channel estimation schemes to illustrate the effect of the channel conditions on the results obtained and only focused on varying the distance of the users from the BS to study the interference among users, rather than varying the signal power levels of the users.

The authors of [2,4,21–23] surveyed all proposed NOMA algorithms, including single carrier, multi carrier, power domain (PD), and cognitive radio-NOMA (CR-NOMA) as well as SISO and

multiple-input-multiple-output (MIMO) NOMA. The main difference between NOMA and MIMO systems is that in MIMO, the system uses multiple antennas to transmit and receive the signals, while NOMA combines the signals as an SISO system with a single transmitting and receiving antenna [3]. The authors in [24] studied the MIMO NOMA system from the single carrier's point of view; the authors compared the BER performance and the power efficiency of the proposed NOMA model in downlink transmission to the theoretical NOMA and OMA performance. This results in a high-power efficiency system as well as an improved system performance. The authors of [2] discussed the modulation and coding schemes that could be proposed in future NOMA systems as well as the security matters in NOMA and the applications of NOMA in future radio generations. The authors in [25] studied the performance of cooperative modulation-based NOMA, where the authors proposed a modulation scheme based on QPSK. The proposed modulation scheme guarantees orthogonality between users through modulation of the near users on the real component and the far users on the imaginary component. The proposed modulation scheme results in a non-complex high-performance system in terms of the BER and the SER.

The performance of deep learning (DL) was comprehensively studied in [24], where the authors proposed an algorithm based on the LSTM network to determine the channel characteristics automatically by training the network with simulated channel data. The results [24] showed an improvement in the sum rates and bit error rates (BERs) when machine learning was utilized compared with conventional NOMA. DL algorithms were also proposed in [16,17,25], where the authors applied and verified the proposed algorithms in traffic control. In [26], the authors proposed a partially linear beamforming filter that could be used to receive and decode the received NOMA signal instead of the SIC algorithm. This adaptive filter exhibited high resolution and robustness against variations in the channel environment. The simulations in [26] showed that this method outperforms both SIC-based detection and nonlinear adaptive filtering in a dynamic wireless environment. Unicast-multicast NOMA systems were studied in [27], where the authors were able to attain diversity orders identical to the number of users in the network.

The objective of this paper is to present a novel PD-NOMA system. Firstly, we propose an effective channel estimation algorithm based on LSTM neural networks, which can dynamically adapt to the behavior of the fluctuating channel conditions. LSTMs are used to investigate the complex channel features of PD-NOMA. LSTM is preferred over traditional RNNs to model long-term dependencies between samples, since traditional RNNs using back-propagation suffer from the vanishing gradient problem and slow learning [28]. LSTM in combination with PD-NOMA can be used to improve the outage probability and user data rates of the conventional PD-NOMA system. Then, a novel power allocation algorithm based on Pascal's triangle is introduced. Our proposed power allocation algorithm is used to distribute the power coefficients amongst N users by assigning the power coefficients according to each user's channel conditions. In addition, we introduce an adaptive coding scheme based on Bose–Chaudhuri–Hocquenghem (BCH) codes due to its superior ability to identify and correct errors. Thanks to this coding scheme, the code rates can adapt to the predicted channel state depending on the SNR values of the users to enhance the error performance of the system in terms of the BERs of users. The aforementioned adaptive coding scheme was initially introduced in MIMO systems in [29]. However, it has never been tested in combination with NOMA. Finally, an adaptive modulation scheme based on constellation rotation and cyclic Q-delay is proposed to overcome channel fading and hence restore the transmitted sequences with fewer errors. The adaptive modulation scheme was previously implemented in digital video broadcasting (DVB) systems [30–32]. The proposed algorithms are discussed with respect to the conventional uplink PD-NOMA system. The system proposed is investigated in terms of BER, outage probability, and sum rates under different conditions.

This paper is organized as follows: First, a detailed description of the conventional PD-NOMA system model is given in Section 2. Second, adaptive coding, cyclic Q-delay, constellation rotation, the proposed LSTM channel estimation algorithm, and the power allocation algorithm are explained in detail in Section 3. Third, the system evaluation is proposed in Section 4. Section 5 summarizes the

simulation parameters and the results of the proposed system evaluation. Finally, Section 6 concludes the paper.

## 2. NOMA System Model

The main idea of PD-NOMA is the efficient utilization of the power domain for multiple access to simultaneously serve several users in the same time slot, frequency band, and with the same code. NOMA assigns larger power to users with lower SNR values, with the purpose of providing a balanced distribution of resources. As shown in Figure 1a, OFDMA divides the available RBs (spectrum) among the available users while keeping the transmitted powers at a constant value. On the other hand, NOMA uses all the available RBs (spectrum) by assigning a different power value to each user, as shown in Figure 1b. Additionally, in OFDMA, the subcarrier separation must follow a certain frequency constraint in order to attain orthogonality. In NOMA, this separation is not applicable as all the bandwidth is utilized by all users [8,33].

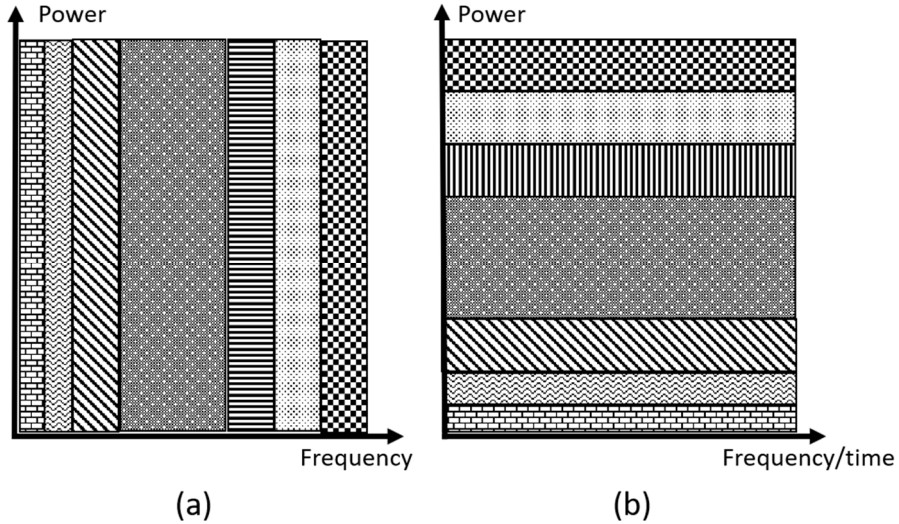

**Figure 1.** Spectrum sharing and power allocation in (**a**) orthogonal frequency division multiple access (OFDMA) and (**b**) non-orthogonal multiple access (NOMA).

The concept of NOMA is based on applying SC at the transmitter and SIC at the receiver, as shown in Figure 2. To transmit the data in NOMA, each user sends an independent pilot sequence at the same time slot for the sake of channel estimation. At the transmitter side, all the separate users' sequences are superimposed into a single signal.

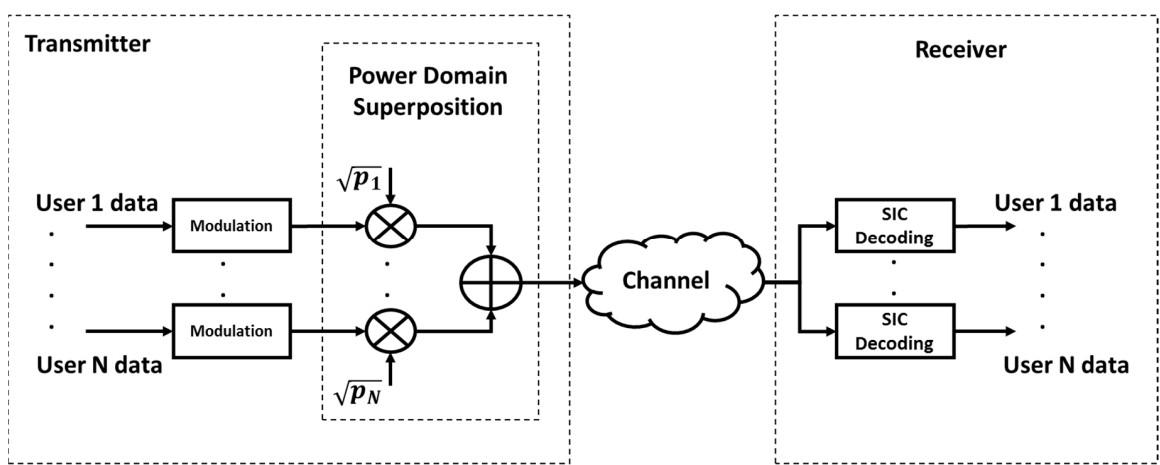

**Figure 2.** Block diagram of the conventional NOMA system.

In NOMA, the transmitter applies superposition coding to superimpose the waveforms of all users and then allocates more power to the user with a lower SNR (worse channel conditions), which is far from the transmitter, and allocates less power to the user with higher SNR (better channel conditions), which is closer to the transmitter. At the receiver terminals, each user uses SIC to decode and receive their own signal, since all users receive the same superimposed signal holding data. This process is performed iteratively until the weakest user receives their own clean signal [2]. The user located near the transmitter cancels the signals of the far users as they act as interference due to their weak SNRs. On the other hand, the farthest user, which has the highest allocated power and, therefore, the maximum contribution to the received superimposed signal, will initially decode and receive their signal [21]. The main challenge for the transmitter is to correctly assign the power for each user according to the channel conditions, as this affects the performance of the SIC [21,34].

*2.1. Superposition Coding (SC)*

NOMA uses SC to transmit the data; SC can encode a message of a low channel condition user at a low data rate, and then superimpose the message of a user with better channel conditions on it. SC is considered one of the essential structures in coding systems due to its ability to attain the capacity region of binary channels (BCs) and Gaussian binary channels [2,11,35].

The BS transmits the superimposed message of $N$ users after assigning the power coefficients as:

$$s = \sum_{i=1}^{N} \sqrt{p_i}\, x_i, \tag{1}$$

where $N$ is the total amount of user equipment (UE), $p_i$ is the allocated power coefficient of user $i$, $x_i$ is the coded and modulated pilot data of user $i$, and $s$ is the superimposed transmitted signal.

Initially, each UE is assigned a random power coefficient under the condition:

$$\sum_{i=1}^{N} p_i = 1. \tag{2}$$

Algorithm 1 provides fair power assignment in conventional PD-NOMA. In this algorithm, users with a low SNR are allocated more power (high power coefficients) than those with a high SNR (better channel conditions).

---

**Algorithm 1** SNR comparison and Power Re-Allocation for UEs

---

**Inputs:** Channel Matrices $(H_1, H_2, \ldots H_N)$ for each UE from the stage of channel estimation (CE)
1: **Loop** I = 1: N
2: $\qquad\qquad\qquad SNR_i = |H_i|^2$ (3)
3: **End loop**
4: Sort SNR values in ascending order
5: Assign the power coefficients according to the SNR values subject to $\sum_{i=1}^{N} p_i = 1$ given by Equation (2).
6: Sort the power coefficients, $p$, in descending order.
7: Update and retransmit the super-positioned sequences with the new power coefficients.
**Outputs:** Power coefficients $(p_1, p_2, \ldots, p_N)$ re-assigned for each user's transmitted message

---

The new power coefficients (fraction of total power) are assigned to the users after each stage of channel estimation, such that their summation is always equal to the total available power at the BS. In previous work, the power coefficients were adaptively chosen, for example, in [20,36,37].

*2.2. Data Transmission*

The signal, $s$, is transmitted through independent Rayleigh fading channels, $H_i$, corresponding to each user, and the channels are contaminated with additive white Gaussian noise (AWGN), $w_i$,

with a power spectral density (PSD) of $N_{0_i}$. The received signal, $y_i$, at the $UE_i$ terminal can be described by the following equation:

$$y_i = H_i s + w_i. \tag{4}$$

### 2.3. Successive Interference Cancellation (SIC)

In SIC, the user with the highest SNR subtracts their message from the super-positioned signal. The same process is followed by the second strongest user. This process is repeated iteratively until the desired signal is decoded [8,28,34]. The SIC method works in an iterative way, imposing lower hardware complexity at the receiver in comparison to conventional multi-user detection schemes. It has been verified that the SIC algorithm can be used to accomplish the capacity regions of the BCs [2]. It has been suggested that the network capacity can be substantially improved with the aid of efficient interference management; hence, SIC is regarded as a promising IC technique in wireless networks [3]. SIC is implemented at the receiver based on the UE channel conditions. Each UE sends pilot data to the BS; pilots are used in the channel estimation (CE) and SNR prediction processes [2,35]. The decoding process is summarized in Figure 3.

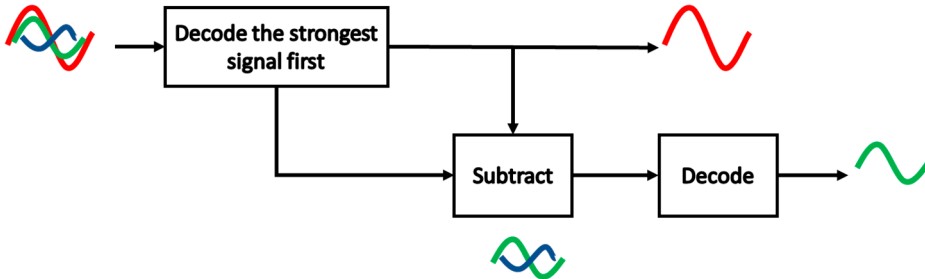

**Figure 3.** Successive interference cancellation (SIC) algorithm [6].

## 3. Proposed System Model

In this section, we discuss the proposed PD-NOMA system model, where a single BS and $N$ UEs are considered. The pilot data $(x_i)$ are generated for each UE independently and then adaptively coded by applying BCH codes. The BCH code rates are randomly initialized and updated after each stage of channel estimation. The coded data $(x_{ci})$ are applied to the modulator input, and the resulting constellation is rotated. The constellation rotation is followed by introducing the cyclic-Q delay. The output data streams, $xq_i$, for $N$ users are initially allocated random power coefficients and are superimposed to determine the signal, $s$, to be transmitted. The signal, $s$, is transmitted by a single BS antenna through $N$ Rayleigh channels. The receiver at each UE terminal applies LSTM channel estimation to determine the channel conditions and the corresponding SNR value for its particular channel. Accordingly, the power coefficients and the BCH code rates are re-assigned depending on the predicted SNR value. Each receiver performs SIC to determine its own signal from the superimposed signal. The complete system model is shown in Figure 4, and the subsequent sections discuss the operation of each individual system component in detail.

### 3.1. Adaptive BCH Codes

The pilot data are adaptively coded by applying BCH codes. BCH codes use special values of $n$ and $k$, where $n$ is the codeword block length and is typically an integer with a value of $2^m - 1$ ($m$ is an integer in the range $3 \leq m \leq 16$). To attain a fairly short BCH code, each power of $m$ must follow some sort of polynomial equation of degree $k$, and $m$ must be the multiplicative order of two modulo $n$ [36,37]. This range of values is available in the form of tables in order to produce any desired BCH code length [38], and $k$ is the number of the information message bits, which is a positive integer given by $k \geq n - m^*t$ ($t$ is the number of errors that could be corrected by the code) [37]. The minimum Hamming distance for a BCH code is $d_{min} \geq 2t + 1$ [39–41]. However, only a few *[n, k]* pair values are

valid [36,42,43]. A BCH code can correct any group of *t* or less errors in a block of *n* bits. The generator polynomial of the BCH code is defined as a function of its solution over the Galois field (GF) *(2$^m$)* [36,44]. By performing the same procedure as in [39], we can determine the generator matrix, parity check matrix, and the equivalent $d_{min}$. Using an [*n*, *k*, *d*] BCH code corresponding to the length of the pilot data, the code rates are varied efficiently along with the channel conditions. This resourcefully reduces the bandwidth consumption, since conventionally, the used BCH code rates range between (63,24) and (255,123) [45,46]. When we have a high SNR value, fewer parity bits are used, and the information bit rate increases. When we have a low SNR value, a lower information bit rate is used, as all the parity bits are transmitted [36,39].

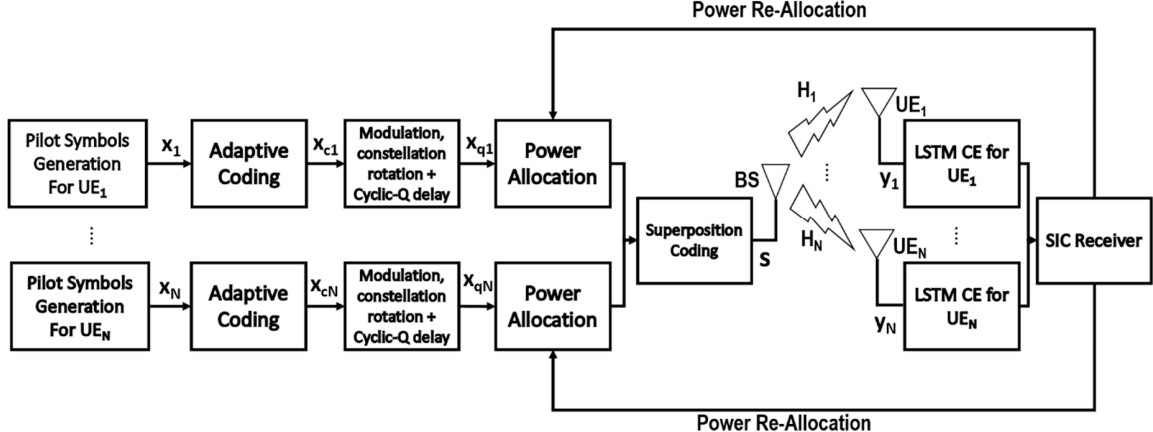

**Figure 4.** Block diagram of the proposed NOMA system.

### 3.2. Modulation, Constellation Rotation, and Cyclic-Q Delay

The coded data are then modulated, and the resulting constellation is rotated. Cyclic Q-delay is introduced to de-correlate the in-phase (I) and quadrature (Q) components to overcome fading in the channel before the superposition coding and transmission stages.

Data are modulated using two schemes, QPSK and 64-QAM. For QPSK, the modulated signal is represented by:

$$x(t) = A\cos(2\pi f_c t + \theta_n), \tag{5}$$

where $0 \le t \le T_{sym}$, A is the amplitude, $f_c$ is the carrier frequency, and $n = 1, 2, 3, 4$.

Then:

$$\theta_n = (2n - 1)\frac{\pi}{4}. \tag{6}$$

For QAM, the modulated signal is given by:

$$x(t) = A_I \cos(2\pi f_c t) - A_Q \sin(2\pi f_c t), \tag{7}$$

where $0 \le t \le T_{sym}$ and $A_I$ and $A_Q$ are the in-phase and quadrature amplitudes and are represented by $log_2(M)$ levels, having the values $-\left(\sqrt{M}-1\right)d$, $-\left(\sqrt{M}-3\right)d$, ... $\left(\sqrt{M}-3\right)d$, $\left(\sqrt{M}-1\right)d$

Where:

$$d = \frac{minimum\ distance\ between\ two\ symbols}{2} = \sqrt{\frac{3(log_2 M)E_b}{2(M-1)}}. \tag{8}$$

The rotated constellation is a technique to decrease the fading effects; this is obtained by adding redundancy with a suitable reference angle, combined with interleaving the symbol components to be transmitted [40]. The transmitted data in a certain component are uncorrelated with the transmitted data in the other component [41–43]. As a result, the I and Q components of a signal are altered by independent fading components [43,45]. This improves the receiver robustness in situations with deep fading [46,47]. The performance improvement is independent of the fading notches, since the

signal components are transmitted through different durations. Constellation rotation results in an improvement in the whole system's performance. The improvement is a function of the rotation angle ($\theta$), which is dependent on the selected modulation and channel type [47].

Constellation rotation is performed according to the equation given below:

$$r_i = Re(e^{\frac{j2\pi\varphi}{360}} x_{i(k)}) + j\, Im(e^{\frac{j2\pi\varphi}{360}} x_{i(k-1)}, \tag{9}$$

where $i$ is the user index with the value 1, 2, 3 ... $N$. Where $\varphi$ is the rotation angle, $x$ is the coded and modulated pilot data for users 1, 2, 3 ... $N$, $r$ is the resultant rotated constellation, and $k = 1, 2, 3 \ldots$. These equations are applied to each user independently.

In order to obtain the signal-space diversity (SSD), constellation rotation is merged with cyclic-Q delay. Constellation rotation and cyclic-Q delay improve performance, particularly when using high code rates or when the channel is subject to deep fading [48].

The combination of rotated constellation and cyclic Q-delay causes the I and Q components to be transmitted at different frequencies, to different carriers, and at different times. Therefore, if either the I or Q is lost, the other component can be used to recover the information.

After rotating the constellation, we introduced the cyclic Q-delay, and the resulting data can be written as:

$$x_{qi} = Re\big(r_{i(k)}\big) + j\, Im\big(r_{i(k+1)}\big), \tag{10}$$

where $x_{qi}$ is the data after the cyclic Q-delay stage, $r$ is the rotated constellation data, $i$ is the user index, w is the value *1, 2, 3 ... N*, and $k = 1,2,3 \ldots$.

### 3.3. LSTM Channel Estimation

Recurrent neural networks (RNNs) are considered a type of supervised learning algorithm. They can model consecutive information for estimation and recognition. RNNs consist of higher dimensional hidden layers made of artificial neurons with feedback loops containing non-linear dynamics [49]. Consequently, RNNs have two inputs, the current and the recent past sample, as shown in Figure 5, where the recent input is the non-looping input to each neuron and the recent past is the output that loops back into the network.

The hidden layers are able to work as memory for the network state at a certain instant, which is conditioned on its preceding state [50]. This construction permits the RNNs to save, recall, and process the previous complex data for an extended period of time [50]. Also, RNNs have the ability to map a certain input to the output sequence during the present time period and forecast this sequence during the following periods of time [51]. Dispersion of the transmitted signal through a fading channel results in an expanded signal with long-term dependencies in between its samples. These dependencies vary from one signal to another and do not follow a certain pattern [28]. Using a feed-forward (FF) neural network to model long-term dependencies will require a high-dimensional feature space with a large number of neurons, leading to over-fitting and sub-optimality [52]. On the other hand, RNNs can model time dependencies with a much smaller number of neurons, due to their feedback recurrent connections. Since, the fading channel is a finite impulse response (FIR) system with a very long impulse response, it may be converted into an infinite impulse response (IIR) filter with a much smaller model and a few properly modeled poles [53,54]. This is equivalent to transforming an FF neural network to an RNN [52].

The traditional RNN using backpropagation through time suffers from the vanishing gradient problem and slow learning [52]. In order to resolve this limitation, Sepp Hochreiter and Jürgen Schmidhuber proposed long-short term memory (LSTM) neural networks [55], since LSTMs are able to model selective dependencies between different portions of the received signal with a small number of neurons and without learning problems. Moreover, they accept vector-based sequence data (where each time step has a vector of measurements), thus incorporating the magnitude and phase parts of the received signal simultaneously [28,52]. LSTMs introduce memory cells, which are able to



store and access data over longer periods of time through their distinct structural design, and preserve the error from back-propagating, unlike the conventional RNN [56,57]. Furthermore, LSTMs are able to efficiently utilize the past time series input data and handle the long-term dependency of time series. The basic LSTM includes an input layer, a hidden layer, and an output layer. Unlike the RNN, the hidden layer of the LSTM contains different gate units that control how the data propagates [58,59].

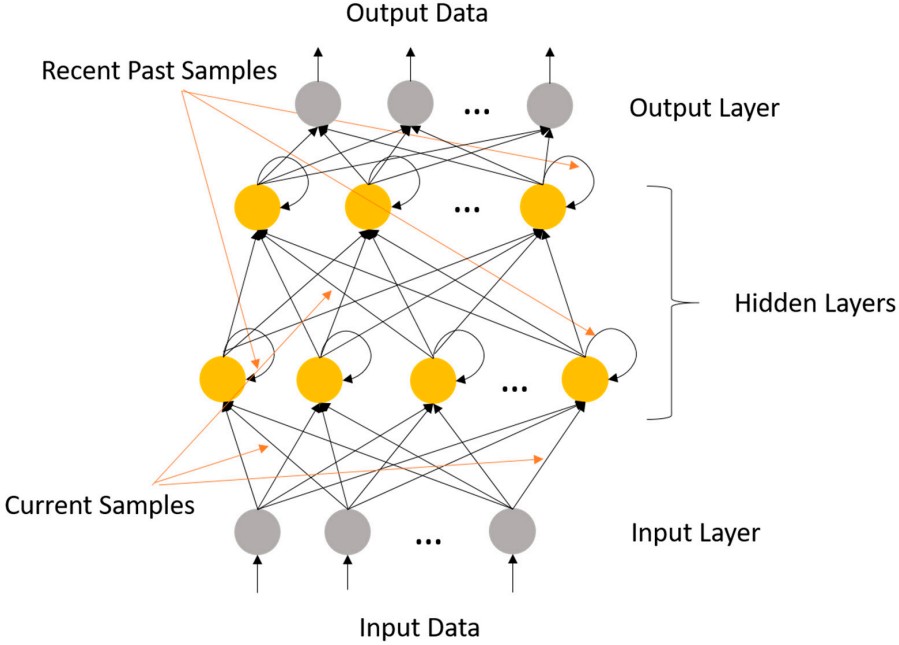

**Figure 5.** Construction of recurrent neural networks.

Figure 6 shows the adopted LSTM channel estimation model, where $t$ is the time-instant (step), $y_{ti}$ is the input, $H_{ti}$ is the output for user $i$ at time instant $t$, and $C_t$ is the memory cell, which is transferred from a hidden layer to the next during each iteration. In addition, a forget gate, input (update) gate, and output gate are added to each hidden layer. The input gate uses the sigmoid function to decide which values are updated, and the output of the input gate is combined with the new candidate values, $\widetilde{C}_t$, from the tan sigmoid (*tanh*) layer to update each state.

The proposed LSTM channel estimation is given by Algorithm 2 below:

---

**Algorithm 2** LSTM Channel Estimation scheme

---

**Inputs:** Transmitted pilot data, $x_i$, for users, target channel matrices, $H_i$.
1: Randomly initialize the weights ($W$'s) and bias ($b$'s) values
2: The forget gate: $f_t = sigmoid\left(\sum_{t=1}^{2000} W_f H_{t-1} + W_f x_t + b_f\right)$
3: The input gate: $i_t = sigmoid\left(\sum_{t=1}^{2000} W_i H_{t-1} + W_i x_t + b_i\right)$
4: The candidate value: $\widetilde{C}_t = tanh\left(\sum_{t=1}^{2000} W_c H_{t-1} + W_c x_t + b_c\right)$
5: Update the old cell state, $C_{t-1}$, into the new cell state, $C_t$, by
$$C_t = (C_{t-1} \times f_t) + \left(i_t \times \widetilde{C}_t\right)$$
6: Update the output of the LSTM by:
   The output gate: $o_t = sigmoid\left(\sum_{t=1}^{2000} W_o H_{t-1} + W_o x_t + b_o\right)$
   The estimated channel matrix: $H_t = o_t \times \tan h(C_t)$
**Outputs:** Estimated channel matrices, $H_t$, and the equivalent SNR values.

---

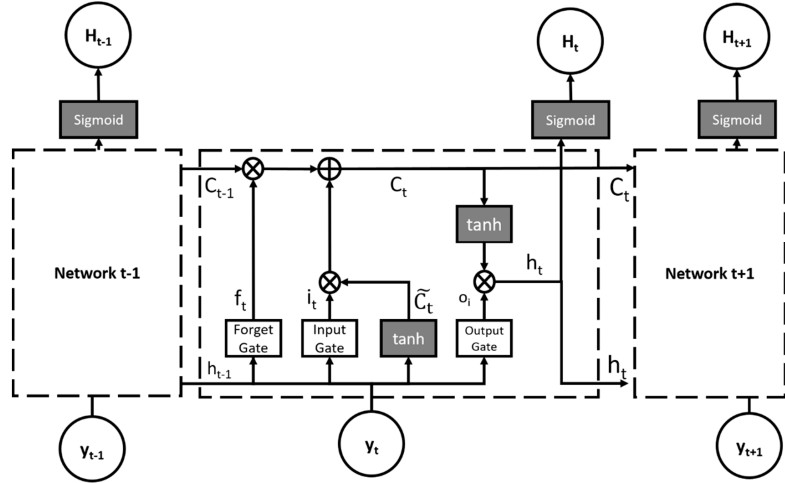

**Figure 6.** Adopted long-short-term memory (LSTM) neural network construction.

### 3.4. Assignment of Power Coefficients

The allocation of power coefficients, as detailed in the literature, can be performed by using different methods based on the users' channel conditions. However, most of these methods have been applied to only two users [1]. In order to adapt to more users, the proposed system is based on Pascal's triangle for power coefficient allocation, which is illustrated in Equation (3). Pascal's triangle is named after the well-known French mathematician and philosopher, Blaise Pascal [60]. It is a triangular array based on binomial expansion coefficients. Pascal's triangle can be applied in different areas in engineering and algebra [60].

Generally, the binomial expansion is formulated as follows:

$$(x+y)^n = a_0 x^n + a_1 x^{n-1} y + a_2 x^{n-2} y^2 + \ldots + a_{n-1} x y^{n-1} + a_n y^n, \tag{11}$$

where $x + y$ is the binomial, and $a_i$ is element $i$ on row $n$ of Pascal's triangle and is equal to:

$$a_i = \binom{n}{i} = \frac{n!}{i!(n-i)!}. \tag{12}$$

In the power allocation algorithm, illustrated by Algorithm 3, the number of users is equivalent to the row number, $n$, under the constraint illustrated by Equation (2).

---

**Algorithm 3** Power Coefficient Allocation

---

**Inputs:** Required number of users, N.
1: **loop** i = 1: N
2: 　　　　$p(i) = \frac{N!}{i!(N-i)!}$
3: **end loop**
4: Sum = $\sum_{i=1}^{N} p(i)$
5: **loop** I = 1: N
6: 　　　　$p(i) = \frac{a(i)}{Sum}$
7: **end loop**
8: **loop** I = 1: N/2
9: 　　　　$p(i) = p(i) \times (4/5)$
10: **end loop**
11: **loop** I = (N/2) + 1: N
12: 　　　　$p(i) = p(i) \times (1/5)$
13: **end loop**
**Outputs:** Power coefficient, $p_i$, for each user.

---

## 4. Proposed System Evaluation

In the proposed downlink NOMA system, the transmitter and the receiver should have complete knowledge of the pilot sequences ($x_i$). The total number of subcarriers and the NOMA system parameters used in the simulations are based upon the long-term evolution (LTE) physical layer standards. The subcarrier spacing should be larger than 10 kHz to reduce performance degradation due to phase noise. Additionally, the subcarrier spacing should lie between 9 and 17 kHz in order to support users up to velocities of 350 km/h. Consequently, a subcarrier spacing of 15 kHz was chosen, which is similar to the case of LTE systems. Moreover, reducing the subcarrier spacing increases the spectral efficiency, as a larger number of data symbols is available for a certain bandwidth. Smaller subcarrier spacing correspondingly guarantees that the fading on every subcarrier is non-selective or nearly flat [31]. All the system parameters used are summarized in Table 1.

**Table 1.** NOMA system parameters.

| Parameter | Value |
|---|---|
| Carrier frequency | 2 GHz |
| Base station (BS) power | 46 dBm |
| System bandwidth (BW) | 5–10 MHz |
| Number of users per cell *(N)* | 10–20 |
| Bandwidth per user | 5.4 MHz |
| Number of data subcarriers | 1200 |
| Number of pilot subcarriers ($x_i$) | 4 |
| Number of guard-band subcarriers | 76 |
| Channel matrices ($H_i$) | Rayleigh or Rician fading |
| Subcarrier spacing | 15 kHz |
| Bose–Chaudhuri–Hocquenghem (BCH) code length | [4,7] up to [26,31] |
| Symbol length | Data: 66.67 msec + cyclic prefix: 4.69 msec |
| Modulation | QPSK and 64QAM |
| Constellation rotation angles | 0.506 rad for QPSK 0.150 rad for 64QAM |
| AWGN ($w$) | −10 to 30 dBm |
| Power allocation coefficients *(p)* | 2/3 and 1/3 3/4 and 1/4 4/5 and 1/5 |

The proposed system was evaluated by means of BER, outage probability, and user sum rates. The upcoming sections illustrate how the outage probability and sum rates were evaluated.

### 4.1. Outage Probability Calculation

The outage probability, given by Equation (13), represents the probability that the BS fails to serve the user or fails to fulfil the QoS requirements of the user. This is given by:

$$P_{outage} = P(BER \leq BER_{TH}), \tag{13}$$

where *P(.)* is the probability that the BER of the user falls below a certain threshold, $BER_{TH}$.

### 4.2. Sum Rate Calculation

The sum rate of NOMA is the maximum rate at which data can be sent over a channel of a definite bandwidth in the presence of noise. For a NOMA system with $N$ users, the rate for user $i$ is given by $R_i$, and the sum rate is given by $R_{sum}$, where:

$$R_i = log_2 \left( 1 + \frac{p_i \text{SNR} |H_i|^2}{SNR_i |H_i|^2 \sum_{i+1}^{N} p_i + 1} \right). \tag{14}$$

And:

$$R_{\text{sum}} = \sum_{i=1}^{N-1} \log_2 \left(1 + \frac{p_i}{\sum_{i+1}^{N} a_i + SNR|H_i|^2}\right) + log_2(1 + p_{\text{N}}|H_N|^2\text{SNR}). \tag{15}$$

## 5. Simulations and Discussion

### 5.1. Simulation Settings

This section provides a description of the simulation environment as well as the simulation parameters used. All the simulations were performed using MATLAB R2018b over an Intel i5 3.20 GHz processor. The performance of the proposed system was evaluated by means of the outage probability, the BER, and the user and sum rates at various SNR values.

The proposed NOMA system was simulated based on $N$ UEs and a single BS over 2000 iterations. At the beginning of each set of iterations, the pilot data, $x_p$, were randomly generated and were known at both the transmitter and the receiver for all UEs. Initially, the power coefficients for the users and the BCH code rates were randomly assigned. The pilot sequences, $x_p$, were modulated by QPSK or 64-QAM. The resulting constellation was rotated by an angle of 0.506 rad for QPSK and 0.150 rad for 64-QAM [32], followed by induction of the cyclic Q-delay. The modulated signals were then superimposed using Equation (1). The superimposed message was transmitted through $N$ independent Rayleigh fading channels perturbed by AWGN. At the receiver, every UE performed channel estimation to determine the SNR corresponding to its channel conditions.

The channel estimation algorithm proposed is the LSTM neural network, which uses gradient-based algorithms rather than the backpropagation algorithm. In order to enable the LSTM to correctly predict the desired channel matrices, it needs to be merged with a fully connected layer. A fully connected layer is similar to the conventional multi-layer perceptron (MLP) neural network. The difference is that it uses a SoftMax activation function in the output layer and that each neuron in the former layer is fully connected to each neuron in the subsequent layer. The main purpose of the fully connected layer is to learn the non-linear combinations of the input features. The sum of all output possibilities from the fully connected layer is 1; this can be guaranteed by the use of the SoftMax as an activation function. The SoftMax function takes a vector of random real-numbers and transforms them into a vector of values ranging from zero to one.

The channel taps used to model the Rayleigh wireless channel were based on the following international telecommunications union (ITU) channel models: ITU channel model A for the indoor office, channel model B for the indoor office, channel model A for outdoor/indoor pedestrian, channel model B for outdoor/indoor pedestrian, channel model A for vehicular environment, and channel model B for vehicular environment [61,62]. At the start of each training phase, the initial weights were randomly initialized. Throughout the training period, according to a gradient descent process, the weights of the network and the bias values were adapted. The training performance of the LSTM network was evaluated by means of the root-mean-square error (RMSE) and the loss, which can be described by Equations (17) and (18), respectively:

$$\text{RMSE} = \sqrt{\frac{\sum_{j=1}^{M}\left(h_j - \hat{h}_j\right)^2}{M}}, \tag{16}$$

$$\text{Loss} = \sum_{j=1}^{M}\left(h_j - h_j\right)^2, \tag{17}$$

where $h_j$ is the actual impulse response channel taps, $\hat{h}_j$ is the predicted impulse response, and M is the number of predictions.

The behavior of the RMSE and the loss during the training epochs are shown in Figures 7 and 8, respectively.

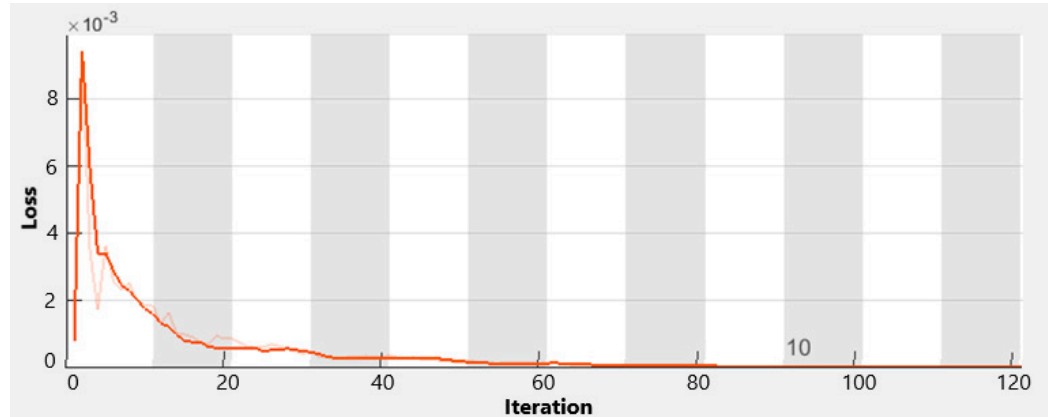

**Figure 7.** Root-mean-square error (RMSE) performance of the proposed LSTM network during the training phase.

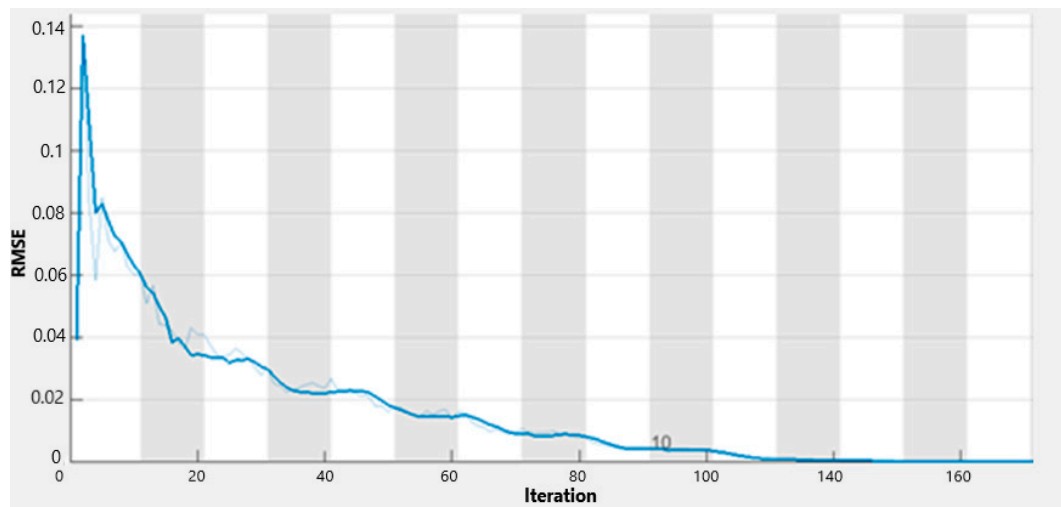

**Figure 8.** Loss performance of the proposed LSTM network during the training phase.

All the parameters used for training and testing the LSTM model are summarized in Table 2. The LSTM channel estimation was tested over 2000 iterations, attaining an average RMSE of $4.12 \times 10^{-06}$ and an average loss of $1.37 \times 10^{-11}$.

**Table 2.** LSTM channel estimation (CE) simulation parameters.

| Parameter | Value |
|---|---|
| Number of layers | 4 |
| Number of neurons/layer (Input layer) | 1000 |
| Number of neurons/layer (LSTM) | 200 |
| Number of neurons/layer (fully connected layer) | 6 |
| Learning rates | 0.005, 0.0001 |
| Length of training data | $1333 \times 1$ |
| Length of testing data | $666 \times 1$ |
| Size of data | $2000 \times 1$ |

*5.2. Simulation Results*

In this section, we provide a detailed description and discussion of the simulation results. Initially, the system was tested under the effect of the LSTM CE without adding the adaptive coding and modulation stages. This set up is indicated as the proposed LSTM NOMA system. To further enhance

the BER as well as the bandwidth consumption, adaptive BCH codes, constellation rotation, and cyclic Q-delay were added to the system. This system configuration is indicated as the proposed adaptive LSTM NOMA system. The results of the CE and the SNR values obtained in the initial step were used to reallocate the power coefficients and select the appropriate code rates for each user. The overall BER of the proposed adaptive system was evaluated and plotted against the SNR.

Figure 9 shows the BER versus the SNR for the proposed PD-NOMA scheme versus the conventional NOMA system and the proposed LSTM-NOMA system without applying the adaptive coding and modulation stages.

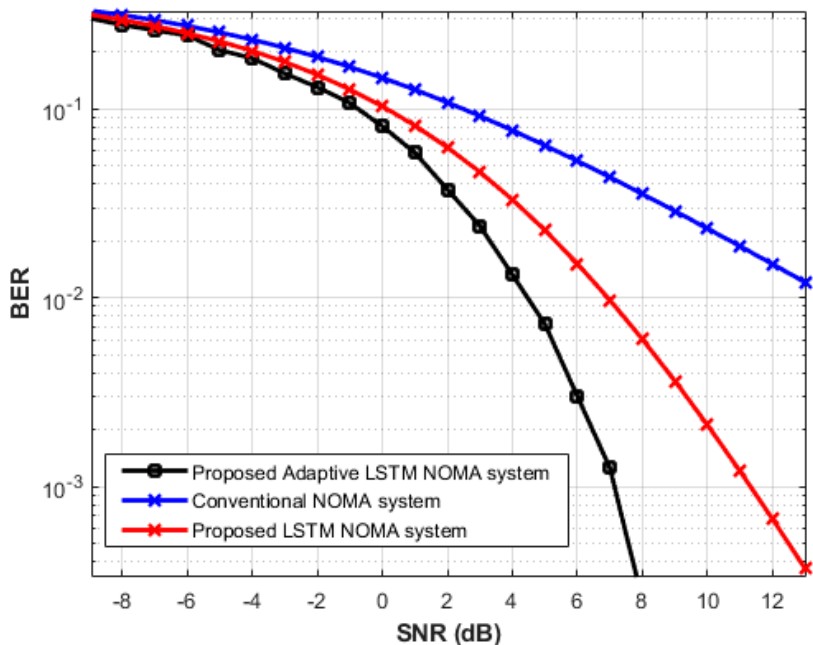

**Figure 9.** Bit error rate (BER) against the signal-to-noise ratio (SNR) for the proposed system versus the conventional NOMA and LSTM NOMA.

The results show that without implementing the constellation rotation and the cyclic Q-Delay, the BER performance reached a minimum of 5% and a maximum of 50% lower BER than that of the conventional NOMA system at SNR values ranging from −2 to 12 dB. In addition, when the adaptive coding and modulation stages were added to the system, the difference reached a minimum of 12% and a maximum of 73% lower BER for the same SNR values. The results verify that supplementing the adaptive coding and modulation stages to the proposed LSTM-NOMA system improves the BER performance and reduces transmission errors when transmitted through a fading channel.

The outage probability of the proposed system was computed against the SNR and is illustrated in Figure 10.

The outage probability for the proposed NOMA system was compared to the conventional NOMA system, and the simulations showed that the proposed system reached an improved outage probability performance for the SNR range of −15 to −2 dB, with a maximum improvement of 10% corresponding to SNR values in the range of −14 to −4 dB. The resulting outage probability for the proposed system was lower than that of the conventional PD-NOMA system as a result of applying LSTM channel estimation, which resulted in a more accurate channel estimate and therefore delivered the specific QoS requirements to the users.

Finally, the user and sum rates of the proposed system were evaluated for two users and compared to the conventional NOMA system.

The plot of the user rates against SNR, shown in Figure 11, indicates that user 1 gained a 33% average increase in the rate, while user 2 gained a 10% average increase in the rate at SNR values

between 9 and 19 dB, when compared to the conventional NOMA system. These enhanced user rates indicate that using the LSTM guaranteed that the users are precisely able to determine the best transmission data rate according to their estimated channel condition.

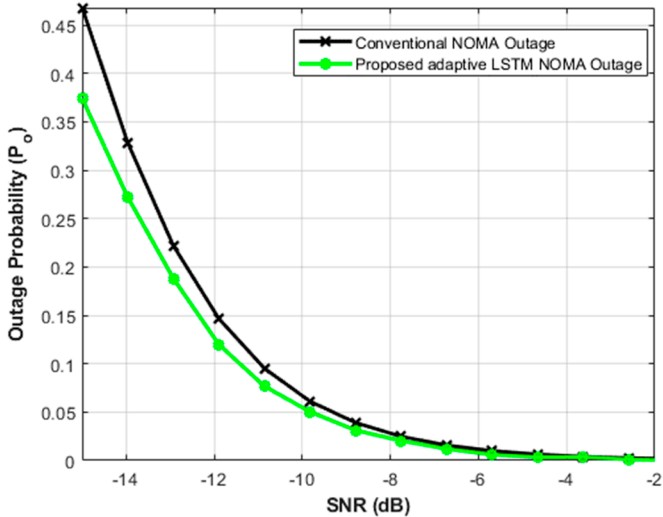

**Figure 10.** Outage probability against the SNR.

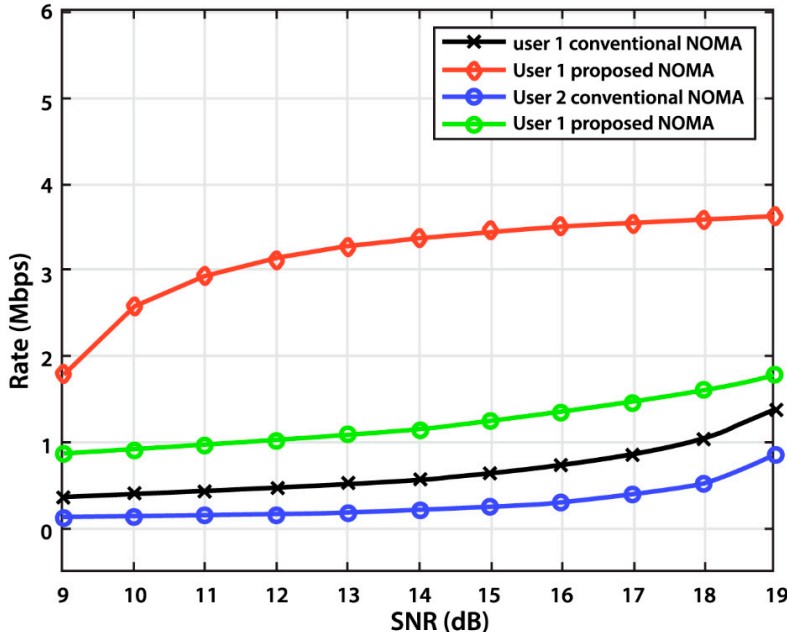

**Figure 11.** User rates for the proposed NOMA system versus the conventional NOMA system for two items of user equipment (UE).

Figure 12 shows that the sum rate of the system increased by an average of 37% at SNR values ranging from 10 to 19 dB when the proposed scheme was applied. The sum and user rates of the proposed system improved due to the fact that the LSTM CE gives a more accurate estimation of each user's channel condition. Therefore, this allows each user to adapt the data rate according to the SNR conditions, attaining the maximum possible transmission quality and, accordingly, the maximum rate.

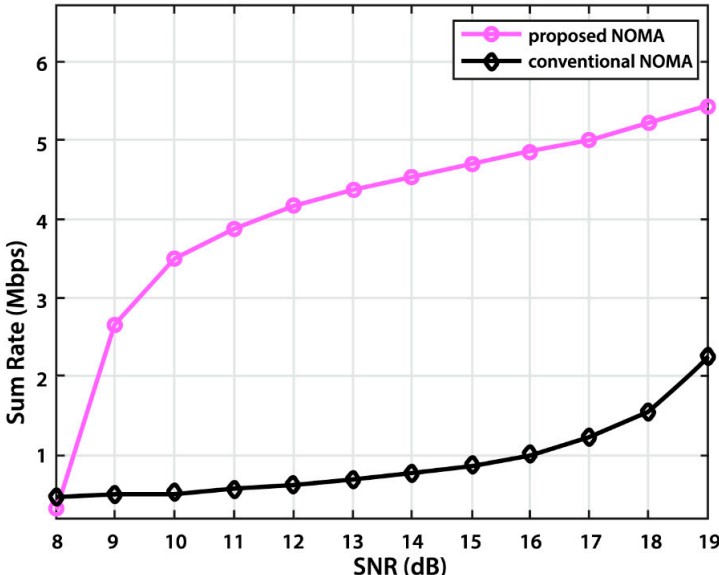

**Figure 12.** Sum rates for two UEs for the proposed NOMA system versus the conventional NOMA system.

## 6. Conclusions

In this work, we presented a new channel estimation technique based on LSTM, aiming to improve the outage probability, BER, and user sum rate of the conventional NOMA system. Furthermore, we proposed a new power coefficient allocation algorithm based on binomial distribution and Pascal's triangle. This new power coefficient allocation algorithm is used to divide power among $N$ users according to each user channel condition. Initially, LSTM channel estimation was performed and the system was evaluated by means of the outage probability, sum rate, and BER. The new NOMA system with the LSTM CE added had a 10% average reduction in the outage probability, a 37% average increase in the sum rate, and a 50% maximum reduction in the BER in the range of −2 to 12 dB in comparison with the conventional NOMA system. To further improve the BER, adaptive BCH codes, constellation rotation, and cyclic Q-delay were added to the new NOMA system. The BCH code rates used were dynamically adapted according to each user channel condition and SNR value. The resulting data were modulated by either QPSK or QAM, and the resulting constellation was rotated. The cyclic Q-delay was applied to the rotated constellation. This added modifications to the new NOMA system, resulting in an additional 23% decrease in the BER in the range of −2 to 12 dB. It is evident from the results obtained that LSTM improves the performance of the NOMA system in terms of outage probability, capacity, BER, and user power allocation accuracy. Furthermore, the BER performance of the system was further improved by adding the proposed adaptive modulation and coding schemes.

**Author Contributions:** Methodology, M.A.; Resources, S.M.G., M.S.E.-M. and A.S.; Software, M.A., M.W.F. and A.S.; Supervision, S.M.G., M.S.E.-M. and M.W.F.; Writing—original draft, M.A.; Writing—review & editing, S.M.G. and M.S.E.-M.

**Funding:** This research received no external funding.

**Conflicts of Interest:** The authors declare no conflict of interest.

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
