# Peer review of "Enhanced NOMA System Using Adaptive Coding and Modulation Based on LSTM Neural Network Channel Estimation"

_applsci, doi:10.3390/app9153022_

Reviewer 1 Report

This paper investigates the performance improving for NOMA system by using LSTM neural network with considering channel estimation. The topic of combing NOMA with machine learning is timely, and the paper is well written. There are some minors:

Before providing Algorithm 1, it would be better if the authors can provide the objective. It seems that Algorithm 1 is used to consider the fairness among users. For fairness and power allocation of NOMA, there are two papers that can be compared in the introduction: [1] Fair non-orthogonal multiple access for visible light communication downlinks. [2] On the optimality of power allocation for NOMA downlinks with individual QoS constraints.

In the abstract, "In NOMA different users are allocated different power" should be "In NOMA different users are allocated with different power".

Author Response

Please find the response (Reviewer1_v4.docx) file attached .

Reviewer 2 Report

Review of manuscript: Enhanced NOMA System Using Adaptive Coding and Modulation Based on LSTM Neural Network Channel Estimation

Main AbdelMoniem et. Al.

This manuscript considers the effective channel estimation (CE) algorithm based on Long-Short Term Memory (LSTM) neural network which can dynamically adapt to the behavior of the fluctuating channel conditions to improve the outage probability and user rates of the conventional NOMA system.  In addition, adaptive coding scheme based on BCH code together with constellation rotation and cyclic Q-delay is introduced to overcome channel fading.

1.      The ideas are not novel however it is for the first time introduced with the combination to NOMA system. In the instruction it mentions that there is power-domain (PD-NOMA) and code-domain (CD-NOMA) but it was not clear if the whole system model is for PD-NOMA. Although one may deduce that the discussion is about PD-NOMA but it would have been much more comprehensive for larger audience if they have stated that the proposed system is for PD-NOMA systems.

2.      The manuscript needs modifications like Figure 1 has low resolution.

3.      Before 2.2 Data Transmission authors mention that adaptively calculate p has not been devised. This is not true, in literature adaptive power coefficients has been devised. It would be good to cite them in this article.

4.      In line 217 it is not clear why m ranges from 3 to 16?

5.      In eq(10) xq_i is the same as x_q shown in 269. It needs to be fixed.

6.       In the Algorithm 2 the symbol \star means convolution operator. Then in the Algorithm 3, the same operator in lines 9 and 12 ? Is it means multiplication?

7.      I wonder if the authors tried to compare the proposed cyclic Q delay with component interleaver that supposed to outperform the former.

8.      The comparisons of BER performance are shown in Figure 1. I wonder if LSTM NOMA system is from the paper “Deep Learning for an Effective Nonorthogonal Multiple Access Scheme” .

Author Response

Please find attached the response file (Reviewer2_v4).docx 

Reviewer 3 Report

A new channel estimation technique based on LSTM,  to improve  the outage probability, BER, and user sum rate against the conventional NOMA system is proposed. Authors also used Nureal network. 

The language of the paper is unacceptable and the figures seems very blur. Authors must follow the standard publication guidelines. 

What channel you used for Fig 9. I think it is AWGN, but the sim results says Rayleigh fading. i think this is misleading. 

The basic details of NOMA is well known, thus, you may curtail. 

The similarity content is 25% and unacceptable. Authors must bring it to 25%.

Authors say, "The Outage probability, given by Equation (16), represents", but I cannot find such equation in 16. 

There are large number of such careless mistakes inherit in the paper. 

Author Response

Please find attached the response file (Reviewer3_v4).docx.

Round  2

Reviewer 2 Report

Thank you for addressing the reviewers comments. The first thing I would recommend is to bring back the quality of images. Whatever method applied to increase the resolution it made it worse and not acceptable. Previous paper was much better.

This manuscript still requires careful revision. As an one obvious example is the citation 35 does not have the location of the conference (e.g., Kaula Lampur). Another problem is in the title of the citation 36 "Optimal Power Allocation Scheme for Non-Orthogonal Multiple
580 Access With $\alpha $ -Fairness" ..and so on. 

Author Response

Please Find Attached Response to Reviewer 2.docx

Reviewer 3 Report

Authors have improved the content based on my previous comments. I have some minor comments :

  In Fig 11, can you present the reasons for your statement "The user rates against SNR, shown in Figure 11, indicate that user 1 gained an average of 33% 442 increase in the rate while user 2 gained an average of 10% increase in the rate at SNR values between."

I think the following papers can be cited to improve the focus and motivation 

Multi-User Detection for the Downlink of NOMA Systems with Multi-Antenna Schemes and Power-Efficient Amplifiers," Physical Communications, https://doi.org/10.1016/j.phycom.2019.01.003

3. Also, add a note in the Intro how your paper is difference from the following paper on modulation based NOMA 

R. Khan,  et al  "Modulation Based Non-Orthogonal Multiple Access for 5G Resilient Networks, "   IEEE Globecom Conference (WS), UAE, Dec 2018. 

Author Response

Please find attached Response to Reviewer 3.docx
